# Investigation of Atmospheric Dynamic and Thermodynamic Structures of Typhoon Sinlaku (2020) from High-Resolution Dropsonde and Two-Way Rawinsonde Measurements

**Lihui Liu [1,2], Yi Han [3,4], Yuancai Xia [2], Qiyun Guo [2], Wenhua Gao [3] and Jianping Guo [3,*]**

[1] Xingtai Meteorological Bureau, CMA, Xingtai 054000, China; yuanfang@cma.gov.cn
[2] Meteorological Observation Centre, CMA, Beijing 100081, China; xumy@cma.gov.cn (Y.X.); gqyaoc@cma.gov.cn (Q.G.)
[3] State Key Laboratory of Severe Weather, Chinese Academy of Meteorological Sciences, Beijing 100081, China; hanyi191@mails.ucas.ac.cn (Y.H.); whgao@cma.gov.cn (W.G.)
[4] College of Earth Sciences, University of Chinese Academy of Sciences, Beijing 100049, China
[*] Correspondence: jpguo@cma.gov.cn; Tel.: +86-10-58993189

**Abstract:** Profiling the vertical atmospheric structure for typhoons remains challenging. Here, the atmospheric dynamic and thermodynamic structures were investigated during the passage of Typhoon Sinlaku (2020) over Xisha Islands in the South China Sea for the period 28 July to 2 August 2020, mainly based on two-way rawinsonde and dropsonde measurements in combination with surface-based automatic weather station observations, disdrometer measurements, and Himawari-8 geostationary satellite images. The study period was divided to three stages: the formation stage of tropical depression (pre-TD), tropical depression (TD), and tropical storm (TS). The wind speed and local vertical wind shear reached the maximum value at 3 km above mean sea level (AMSL) before the typhoon approached the Xisha islands. Pseudo-equivalent potential temperature ($\theta se$) was found to decrease with the altitude below 2 km AMSL; temperature inversions occurred frequently within this altitude range, particularly during the TS stage. This seemed a typical capping inversion that indicated a downward motion above 2 km AMSL. The temperature increased slightly with the development of Typhoon Sinlaku (2020) at altitudes of 8–10 km AMSL. This indicated that our observations presumably captured the air mass warmed by the condensation, which was a good signature of an upper air in the tropical cyclone. In addition, wind speed (particularly in the lower stratosphere), specific humidity, and equivalent potential temperature escalated significantly when the tropical depression strengthened into Typhoon Sinlaku (2020), which indicated that the typhoon constantly obtained energy from the sea surface during its passage over the study region. The thermodynamic and dynamic structures of atmosphere advance our understanding of the inner structure of typhoons during the different evolutionary stages.

**Keywords:** dropsonde; profile; wind shear; typhoon; South China Sea

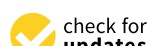



## 1. Introduction

A tropical cyclone (TC) is one of the most devastating natural phenomena in the world. It takes in such forms as tropical depression, tropical storm, and typhoon. It always initiates in warm atmospheres over a warm ocean [1–3] and may grow into a destructive weather system, with strong winds and heavy rainfalls [4,5]. China is one of the countries most affected by TCs in the world, with an average of 9.3 TCs each year [6,7]. TCs are the most serious natural disasters in China, posing a great threat to economic development and to people's lives and property [8,9].

A persistent challenge is the accurate prediction of the track and the intensity of TCs [10,11]. With the development of satellite and numerical modelling, the track prediction of TCs has been improved significantly [12,13], whereas intensity forecasting still

has a relatively large uncertainty [14]. Halverson et al. [15] pointed out that the lack of understanding of the dynamic and thermodynamic structures of TCs and their evolutionary processes were the key factors limiting the improvement of the forecasting ability of the intensity of the TCs. Therefore, research related to the internal structure of TCs has received increasing attention in recent years. Generally speaking, the TCs spend more than half of their entire lifetimes over the ocean; thus, the study of TCs before landing is of great importance [16,17]. However, due to the limited number of meteorological observing platforms and buoys, few studies involved in characterizing internal dynamic and thermodynamic structures of TCs were performed near the coastal regions of eastern Asian countries like China and Japan [18,19].

As early as 1977, Frank et al. [20] used 10 years of radiosonde anemometer data to describe the typical structural characteristics of typhoons in the Northwest Pacific. They found that mature typhoons had a warm-core structure in the upper layer, and there was a cyclonic (anticyclonic) circulation in the lower-middle (upper) troposphere. Kepert et al. [21] studied the boundary layer wind characteristics of tropical cyclone Georges (1998) and found that the dynamic structure of the boundary layer of the tropical cyclone has obvious asymmetry, and its structure has different characteristics at different stages of tropical cyclone development. Dodge et al. [22] compared the vertical structures of the two strongest hurricanes and found that the similarities and differences in their structures can represent the differences in their subsequent development. In the northwestern Pacific Ocean, Xu et al. [23] used satellite data and high-resolution numerical simulation methods to study the helical structure of typhoon Ute in the South China Sea; other scholars [24–26] studied the thermal dynamics after the typhoon landed. However, due to the lack of actual observational data, the dynamic and thermal structural characteristics of typhoons in the Northwest Pacific is still underexplored.

In recent decades, with the rapid development of aircraft observation, weather radar, and satellite and other observational technologies, more observation data with high spatial and temporal resolution have been extensively used to conduct the TC-related research [27–29], which has helped shed light on the internal structures of TCs over the ocean. For instance, it has been well recognized that both the weak vertical wind shear [30] and the fast movement of asymmetric structure are possible reasons for the strong updraft motion in TC [31,32] and the rapid intensification of TCs when the translation speed is fast [33].

At present, the sounding balloon is one of the ideal instruments for in situ measurements of atmospheric pressure, temperature, and other thermodynamic variables such as turbulence and gravity wave [34,35], which can fill the observational gap over remote ocean areas [36]. In the 1990s, a few dropsondes were applied to study hurricanes. To study the atmospheric thermal and thermodynamic characteristics of Hurricane Bonnie during its landfall in 1998, the dropsondes deployed by WP-3D aircraft by the National Oceanic and Atmospheric Administration (NOAA) were used by Schneider and Barnes [37]. They found that the intensification of the hurricane was probably related to the different characteristics of the onshore flow and offshore flow during the landfall of Hurricane Bonnie, which resulted in the formation of asymmetric dynamic and thermal structures at the bottom of the hurricane. This asymmetry reached its maximum at 10 m above mean sea level (AMSL). In addition, some scholars have focused their attention on the storm environment of typhoons occurring over the western Pacific Ocean using dropsonde measurements, showing significant improvement of forecasting skills [38–40] by considering the dynamic and thermodynamic features of typhoons [41,42]. Nevertheless, few previous studies paid attention to this issue in oceans near mainland China because of the scarcity of sounding observations [43,44].

When validating the dropsonde measurements against the simultaneously measured radiosonde data, the data quality of the dropsonde seems satisfactory [45]. It follows that high-quality dropsonde data collected in the TCs have made a significant contribution to the improved prediction of TC tracking [46–50]. Additionally, the assimilation of even

a subset of dropsonde data from fully sampled target regions can produce a statistically significant reduction in track forecasting errors (up to 25%) within the critical first two days of the forecasting [49,50].

Observational studies with dropsondes have found that the dynamic and thermodynamic condition differed in the eye, eyewall, outer core, and ambient regions of TC [51], and the change of hurricane intensity is highly related to the internal dynamic and thermodynamic structure of the hurricane. Zhang et al. [52] found that the thermodynamic structure of the hurricane boundary layer and the asymmetry of convection were closely related to the change of hurricane intensity when the dropsondes were used to study the dynamic and thermodynamic condition of Hurricane Edouard in 2014. However, in the Western Pacific, where typhoons are present every year, there are only a few studies of typhoons observed by rawinsonde and dropsondes [53,54] that attempt to elucidate the internal structures of typhoon. For instance, ten dropsondes were deployed by an aircraft when a typhoon was approaching Hong Kong, China, in 2016, and some unique structural characteristics of TC were revealed in the research [54]. It was found that both the ensemble-mean profiles of dry-bulb temperature and potential temperature were linearly distributed along altitude. More interestingly, the profiles of the TC's pressure deficit were linear along barometric altitude, with the slope of pressure rising linearly with the increase of the pressure deficit at mean sea level. Although satellite observations have been widely used to study the structures of typhoons, some limitations still exist compared with in situ measurements of typhoon structures [55].

Typhoon Sinlaku (2020), the third tropical storm initiated over the Northwest Pacific Ocean, formed from a tropical depression at 07:00 UTC on 31 July 2020 and then strengthened into a tropical storm at 07:00 UTC on 1 August approximately 20 km west of the coastal city of Sanya, Hainan province of China [56]. During the passage of Typhoon Sinlaku (2020) from 28 July to 2 August 2020, the South China Sea Experiment 2020 of the "Petrel Project" was carried out, during which a series of high-resolution dropsondes and two-way rawinsondes were deployed. To the best of our knowledge, this is the first experiment in recent years conducted by the Chinese government to elucidate the internal structures of a typhoon over the South China Sea. These unique and valuable observations provide us an unprecedented opportunity to analyze the dynamic and thermodynamic structure of Typhoon Sinlaku (2020).

The rest of this paper proceeds as follows: Section 2 presents a brief overview of the data used in this study. Section 3 provides a summary of Typhoon Sinlaku (2020). Section 4 shows the classification of different evolutionary stages of Typhoon Sinlaku (2020) and their corresponding internal structures during its passage over Xisha station (112.33°E, 16.83°N) in the Western Pacific. This study ends with summaries and conclusions in Section 5.

## 2. Materials and Methods

The soundings of atmospheric states (thermodynamic and dynamic characteristics) were mainly acquired from the intelligent, two-way rawinsonde system, which constitutes part of the Comprehensive Marine Observing Experiment under the auspices of the Petrel Meteorological Observation Experiment Project of the China Meteorological Administration [55], as detailed in Table 1. The two-way rawinsonde balloons were launched at the Xisha site, which underwent three stages, the ascent stage, drift stage, and descent stage, as represented by lines in different colors in Figure 1. To see the changes in dynamic and thermodynamic features at different evolutionary stages of Typhoon Sinlaku, the lifetime of this typhoon was divided into three stages: the formation stage of tropical depression (pre-TD), TD, and tropical storm (TS), which is illustrated in Figure 1. It should be noted that the dropsondes aboard the unmanned aerial vehicle (UAV) were deployed on the periphery of the storm (TS stage), while the two-way rawinsondes were deployed to obtain the pre-storm condition (Pre-TD stage) and storm-inner core (TD stage), as shown in Figures 1 and 2. The measurements of the drift and descent stages were used to fill the gap left by operational radiosonde measurements, which were typically conducted twice per day. The measurement

uncertainties of operational radiosonde measurements have been well verified and thus can be reliable enough to be used here [57–61]. Compared with the operational radiosonde data, which were only available for 00:00 and 12:00 UTC, the observational time of the dropsonde seemed more flexible (theoretically, the dropsonde can be deployed at any time), thereby giving us a unique opportunity to characterize the internal structures of the typhoon during its evolutionary stages or at various storm-relative positions.

**Table 1.** Statistics of the intelligent, two-way horizontal drifting radiosonde system deployed from the Xisha Meteorological Observing Station. Note that all times refer to local standard time (LST).

| Date | Ascent Stage | | | Drift Stage | | | Descent Stage | | |
|---|---|---|---|---|---|---|---|---|---|
| | Start Time | Finish Time | Number of Samples | Start Time | Finish Time | Number of Samples | Start Time | Finish Time | Number of Samples |
| 28 July | 10:27 | 11:45 | 4096 | 11:45 | 12:28 | 2573 | 11:51 | 13:09 | 4569 |
| 28 July | 13:11 | 14:36 | 5077 | 14:36 | 15:17 | 2484 | 15:17 | 16:12 | 3268 |
| 31 August | 22:09 | 23:05 | 3340 | | | | | | |
| 1 August | 01:06 | 01:52 | 2811 | | | | 01:59 | 02:24 | 907 |

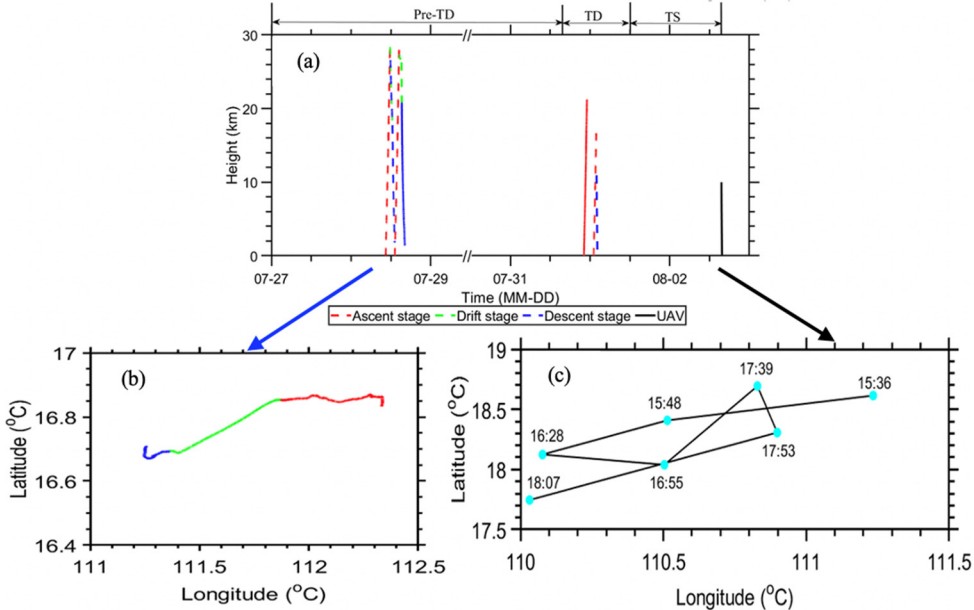

**Figure 1.** (**a**) Time-height cross-section for the two-way rawinsondes and UAV dropsondes launched during the South China Sea Experiment 2020 of the "Petrel Project", designed to characterize the atmospheric environment of Typhoon Sinlaku (2020). The ascending, drifting, and descending stages are denoted by red, green, and blue color. The black line represents the dropsondes deployed by the UAV. The solid lines denote the observations used in this study. The lower left panel (**b**) shows the trajectory of a two-way rawinsonde during pre-TD, and the lower right panel (**c**) shows the track of UAV for flight leg III as listed in Table 2. Note that the time is in local standard time (UTC + 8 h).

**Table 2.** The information on dropsonde measurements conducted from the unmanned aerial vehicle on 2 August 2020. Note that all times are in LST.

| Unmanned Aerial Vehicle | Start Time (hh:mm) | Finish Time (hh:mm) | Number of Dropsondes |
|---|---|---|---|
| Leg I | 15:50 | 18:17 | 8 |
| Leg II | 15:38 | 18:01 | 10 |
| Leg III | 15:36 | 18:10 | 7 |
| Leg IV | 16:57 | 18:04 | 4 |

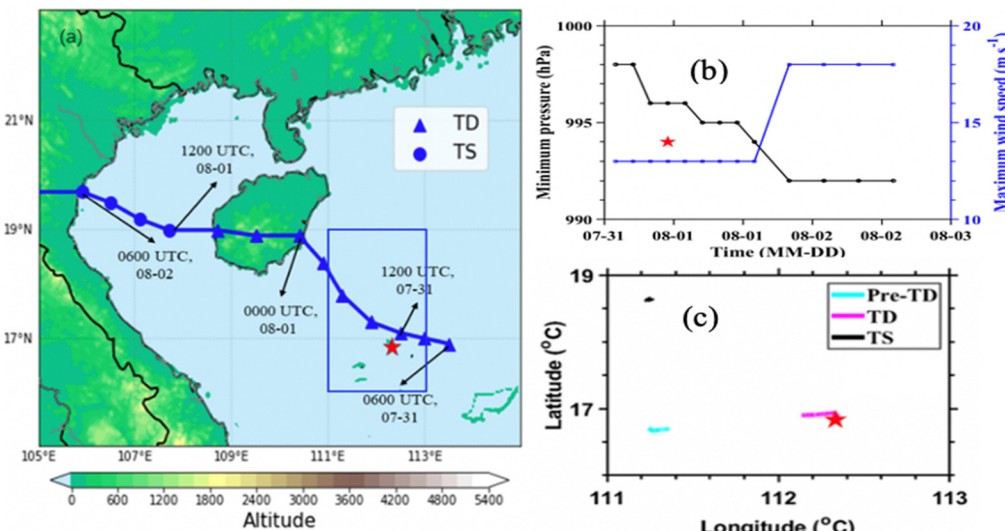

**Figure 2.** The track of Typhoon Sinlaku (2020) passing through the South China Sea, which moved from southeast to northwest; the solid blue triangle represents the location of a tropical depression (TD) and the solid circles represent the locations of tropical storms (TS). The red pentagram in the left panel represents Xisha station (112.33°E, 16.83°N). Also shown are (**b**) the minimal central pressure and maximum wind speed and (**c**) the spatial extent (blue rectangle in panel (**a**) of the trajectory the of the drifting balloon and the dropsondes that correspond to the three different stages of Typhoon Sinlaku (2020) for the study period: pre-TD stage in cyan solid line, TD stage in magenta solid line, and TS stage in black solid line. The red pentagram in the right panel represents the time when the typhoon passes over Xishan station. Note that the time is in UTC.

Figure 1 shows the time-altitude cross-section for the dropsondes launched during the South China Sea Experiment 2020 of the "Petrel Project", which was designed for investigation of the atmospheric environment of Typhoon Sinlaku (2020). Also shown are the trajectories of sounding balloons at three different evolutionary stages. The track of Typhoon Sinlaku (2020) for the period from 06:00 UTC 31 July to 06:00 UTC 2 August 2020 is shown in Figure 2a. The acquisition times of both the GFS reanalysis and Himawari geostationary dataset on 28 July, 31 July, and 2 August 2020 in Figure 3 approximately corresponded to the time range of sounding balloons in the corresponding days shown in Figure 1. As shown in Figure 2c, the trajectories of the operational radiosonde, dropsonde, and two-way drift rawinsonde were constrained to a limited spatial extent surrounding Xisha station, resulting in the observations obtained from this experiment not able to be used for analysis of different portions of the typhoon. Therefore, in this study, we mainly attempted to elucidate the dynamic and thermodynamic structures of Typhoon Sinlaku (2020) at different developing stages.

The dropsonde data from the high-altitude large UAV named Wing Loong were also adopted in this study, which were deployed on the periphery of the storm. On 2 August 2020, a total of 29 dropsondes were deployed by a UAV from 15:36 to 18:17 local standard time (LST), organized into four flight legs (Table 2). The panel in the lower right corner of Figure 1 shows the trajectories of the UAV when the third flight leg of the dropsondes was deployed. The black vertical line in the main panel of Figure 1 refers to a flight leg that corresponded to flight leg III from 15:36 to 15:47 LST. All dropsondes were initially deployed at approximately 300 hPa, and the data on temperature, relative humidity, pressure, and wind were collected during their 10 min descents. To ensure robust and reliable analysis in the following section, these measurements were subject to a thorough quality-control procedure. For instance, outliers were identified and removed by comparison with the data above and below the target points in the vertical direction [62].

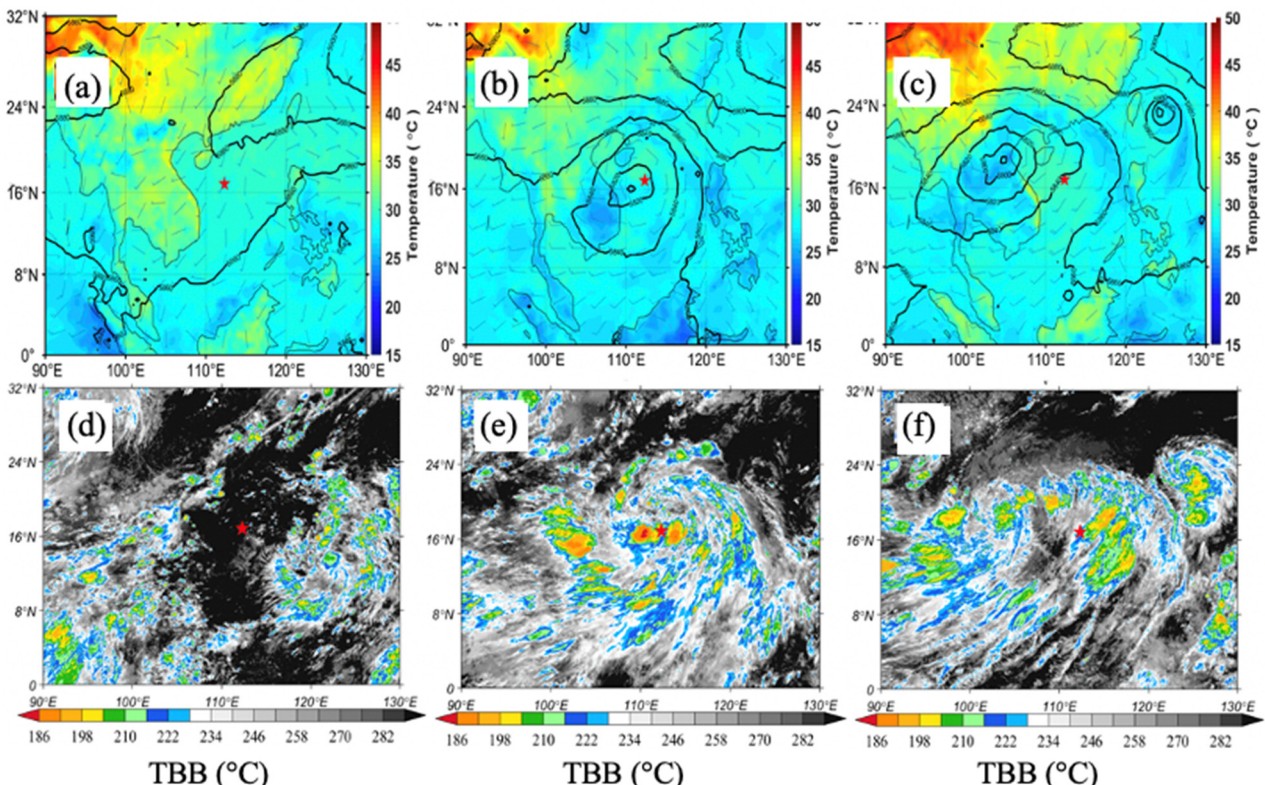

**Figure 3.** (**a–c**) The field of air temperature at the surface (color shading), wind field at 850 hPa (wind barb), and geopotential height at 500 hPa from GFS reanalysis (solid black lines); (**d–f**) the TBB images from the Himawari-8 geostationary satellite acquired at 06:00 UTC on 28 July, at 12:00 UTC on 31 July, and at 12:00 UTC on 2 August 2020, corresponding to three different stages of Typhoon Sinlaku (2020): (**a,d**) pre-TD stage, (**b,e**) TD stage, and (**c,f**) TS stage. The red pentagrams in all panels show the position of Xisha station.

The other meteorological dataset used in our study included the Global Forecast System (GFS) with $0.5 \times 0.5°$ resolution, the black-body brightness temperature (TBB) obtained from the Channel 13 (with a central wavelength at 10.4 μm) of the Advanced Himawari Imager onboard the Himawari-8 geostationary meteorological satellite [63,64], the typhoon best track data from the China Meteorological Administration [65,66], and the surface rain gauge data and disdrometer data at Xisha Station. Based on the aforementioned observations, several geophysical variables related to the typhoon properties were also derived, including vertical wind shear, equivalent potential temperature ($\theta$e), and pseudo-equivalent potential temperature ($\theta$se). All these variables were utilized in the consideration of the dynamic and thermal structure of Typhoon Sinlaku (2020), in combination with the generalized intercept parameter ($N_w$) and the mass-weighted mean diameter ($D_m$) from disdrometer observations at Xisha station.

## 3. Results

The center of Typhoon Sinlaku (2020) was near the Xisha Islands, at about 12:00 UTC 31 July 2020. Then, it swept Hainan Island, moved northwestward to the Beibu Gulf, and finally made landfall in Thanh Hoa, Vietnam, at 06:00 UTC on 2 August with a minimum central pressure of 992 hPa and a maximum wind speed of 18 m s$^{-1}$ [56]. The trajectory is displayed in Figure 2a.

### 3.1. Determination of the Evolutionary Stages of Typhoon Sinlaku (2020)

The measurements from the radiosonde balloons launched during the period from 28 July to 2 August 2020 at the Xisha site covered the entire development of Typhoon

Sinlaku. The pre-TD stage refers to the period from 27 July to 06:00 UTC 31 July 2020. Figure 3d shows that no clouds were present in the study region surrounding Xisha station, which meant that the vertical soundings reflected the pre-storm environment. Particularly, two sounding balloons were launched during this phase, and the measurements from the descending stage of the balloons from 15:17 to 16:12 LST 28 July 2020 were used in this study. At this time, TBB was higher over the ocean surface, and no obvious circulation was formed (Figure 3a).

The TD stage was from 06:00 UTC 31 July to 12:00 UTC 1 August 2020, when the tropical depression formed, and a heavy rainfall event was recorded at Xisha station (Figure 4a). In addition, two balloon launches took place during the experiment. The study used data from the ascending stage of the balloon experiment from 22:09 to 23:05 LST 31 July 2020. The Xisha Islands were near the center of the TC. At this time, the cyclonic circulation of the typhoon was weak, and no obvious spiral convective band or typhoon eye wall was formed (Figure 3b,e).

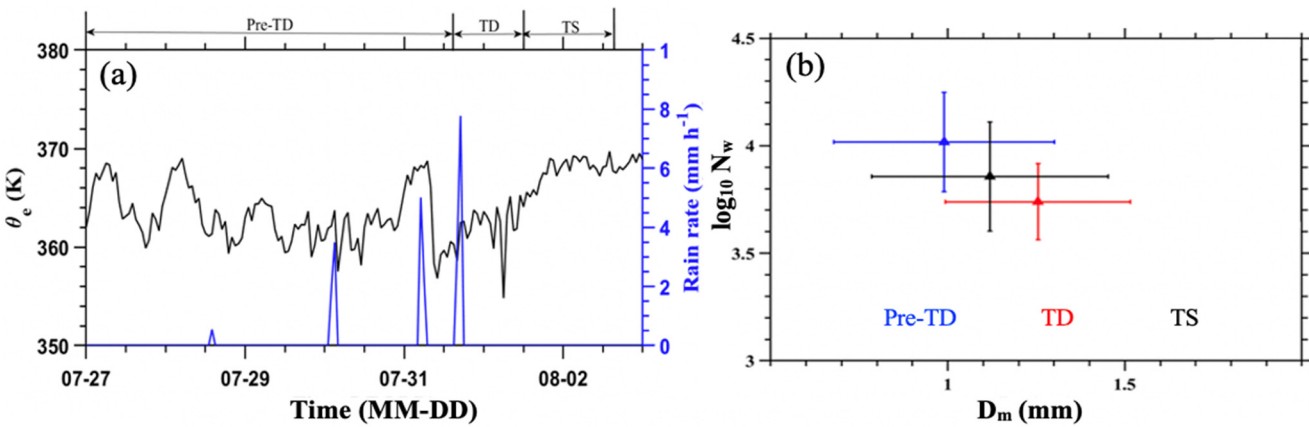

**Figure 4.** (**a**) The time series of $\theta e$ and rain rate, and (**b**) the correlation between $D_m$ and $\log_{10}N_W$ for the raindrop size distribution from disdrometer observations at Xisha station (112.33°E, 16.83°N), colored by three different evolutionary stages of Typhoon Sinlaku (2020). Note that the time is in local standard time (UTC + 8 h).

The period after 12:00 UTC 1 August 2020 was thought of as the TS stage, when the TC intensified into a tropical storm. Four flight legs of dropsonde experiments were deployed. The measurements of dropsondes from 15:36 to 15:47 LST 2 August 2020 in flight leg III were used in this study. From the perspective of large-scale circulation, we can find that the typhoon in the west of the Xisha Islands and the cyclonic circulation of the TC strengthened, which formed a relatively closed typhoon eye wall and spiral convective band (Figure 3 c,f).

### 3.2. Atmospheric Profile Features for Different Evolutionary Stages of Typhoon Sinlaku (2020)

Figure 4a presents the time series of the changes in the equivalent potential temperature ($\theta_e$) and rain rate at the Xisha station. $\theta_e$ showed a generally decreasing trend from the pre-TD stage to the TD stage. Notably, it dipped sharply on 29 July and then reached a minimum of 354 K on 1 August. After the typhoon passed the Xisha station, $\theta_e$ increased at the end of TD period, reaching as high as 368.3 K in the TS stage. The $\theta_e$ observed in the near-storm environment for Typhoon Sinlaku (2020) was almost congruent with 363 K at 0.5 km AMSL, which was similarly observed outside the eye wall of another typhoon [41]. As the typhoon passed by the Xisha site, the rain rate at Xisha station increased, reaching a maximum of 7.7 mm h$^{-1}$ in the TD stage. It is also worth noting that the observed precipitation events recorded by the rain gauge at the Xisha site were always accompanied by a decrease in $\theta_e$, which was largely owing to the downdrafts that brought air with low $\theta e$ values from the troposphere into the near surface [67,68].

Figure 4b showed the changes in the microphysical properties of the raindrop size distribution detected by the disdrometer in different evolutionary stages of Typhoon Sinlaku (2020). Parameters included generalized intercept parameter ($N_w$) in logarithmic form $\log_{10}N_w$ and the mass-weighted mean diameter ($D_m$), which reflected the number concentration and the proportion of large particles in the precipitation, respectively. Following references such as [69], a $D_m$-$\log_{10}N_w$ diagram was utilized to illustrate the raindrop size distribution. We found that $D_m$ first increased and then decreased during the development of the typhoon, which changed from 0.99 mm in the pre-TD period to 1.25 mm in the TD period and ultimately to 1.12 mm in the TS period. In addition, $\log_{10}N_w$ generally showed a first decreasing and then increasing trend. This phenomenon suggested that in different evolutionary stages of this typhoon, the increase the in size of raindrops was always accompanied by the decrease in their number concentration, which was consistent with the conclusions obtained by Wen et al. [70]. In their study, the characteristics of the raindrop size distribution of seven typhoons after making landfall in China were studied. Also noteworthy was that the changes in the raindrop size distribution as revealed in Figure 4b may reflect differences in the distance of the observational site relative to the TC center, except for various evolutionary stages of Typhoon Sinlaku (2020).

Figure 5a–c shows the wind speed profiles detected by the sounding balloons at different evolutionary stages of the typhoon. Overall, during the stages of pre-TD and TD, wind speed tended to increase with height in the lower troposphere up to approximately 9 km AMSL and then decreased with height up to about 11 km AMSL, reaching its maximum value at 17 km AMSL. As expected, the maximum wind speed showed a significant increase from 23.1 m s$^{-1}$ during the pre-TD stage to 28.3 m s$^{-1}$ during the TD stage. In other words, the wind in the stratosphere strengthened as Typhoon Sinlaku (2020) passed over Xisha station. In the lower layer (below 6 km AMSL), the maximum wind speed appeared at around 3 km AMSL, which somehow resembled the conclusions from previous studies [40,44,71]. During the pre-TD and TD stage, the maximum wind speeds in the lower layer were around 9 m s$^{-1}$, while during the TS stage, when the typhoon intensified, the maximum wind speed in the lower layer rose up to nearly 15 m s$^{-1}$. The local vertical wind shears, shown as solid red lines in Figure 5d–f, reached the maximum at around 3 km AMSL in the pre-TD stage, followed by a rapid drop above and below 3 km AMSL. This roughly agreed with Young et al. [72], who reported a similar phenomenon.

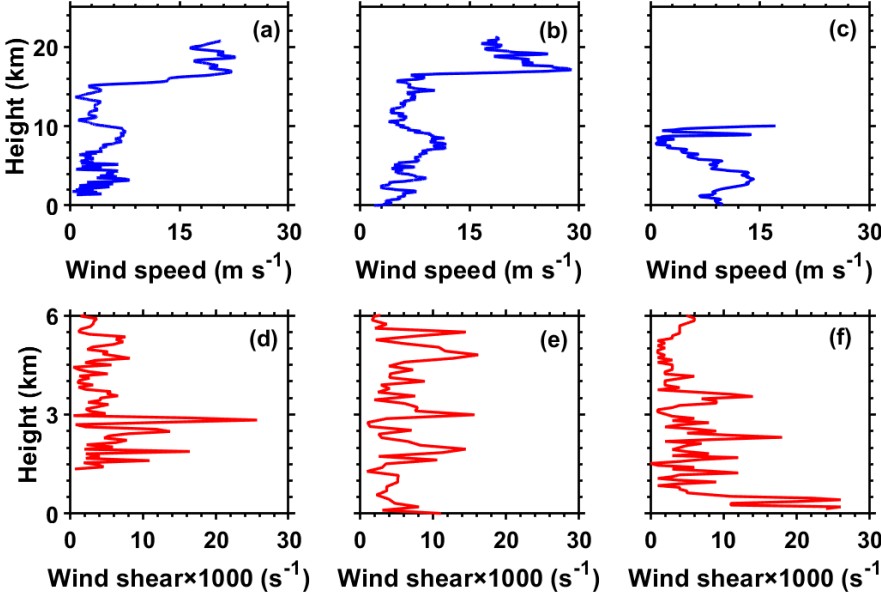

**Figure 5.** The profiles of wind speed (**a**–**c**) and local vertical wind shear (**d**–**f**) from two−way rawinsonde and dropsonde measurements for three different evolutionary stages of Typhoon Sinlaku: pre-TD (**a**,**d**), TD (**b**,**e**), and TS (**c**,**f**).

## 4. Discussion

The atmospheric environmental conditions will be discussed for Typhoon Sinlaku (2020) at various development stages. As illustrated in Figure 5, the atmospheric dynamic structures showed variation over time and height. The role of water vapor by dynamic feedback in circulation has been addressed in previous studies [72,73]. In this section, we will discuss the role of thermodynamic features in the evolution of Typhoon Sinlaku (2020).

Figure 6 shows the vertical profiles of temperature, pseudo equivalent potential temperature (θ$se$), and specific humidity in the different development stages of the typhoon. It is shown in Figure 6a that the temperature increased slightly with the development of the typhoon at altitudes of 8–10 km AMSL, which corresponded to the upper troposphere. This indicated that our observations presumably captured the air warmed by the condensation, which was a good signature of upper air in a tropical cyclone. Overall, the lapse rates in the three stages were similar throughout the atmosphere as measured by soundings. The lapse rate in the period of the TD is slightly smaller than that in the period of the pre-TD and TS, which was first decreasing and then increasing with altitude, as also observed in previous studies [74]. The ensemble-mean atmospheric temperature decreased almost linearly with the increase of altitude below 10 km AMSL. The lapse rate below 10 km AMSL was 6.25 K/km, compared with that of 5.63 K/km. This means that the lapse rate of the temperature in the TD period was slightly higher than that in the pre-TD period above 10 km AMSL, which is consistent with previous studies [75]. Figure 6b,c presents the profiles of $\theta se$ and specific humidity increases with the development of the typhoon. Below 3 km AMSL, the largest difference in $\theta se$ among the different stages reached up to as large as 8 K, while the largest difference of specific humidity reached 0.002 g g$^{-1}$. This indicated that the warm core of the typhoon was continuously strengthened, which resulted in energy being easily transported continuously from the underlying ocean [76,77].

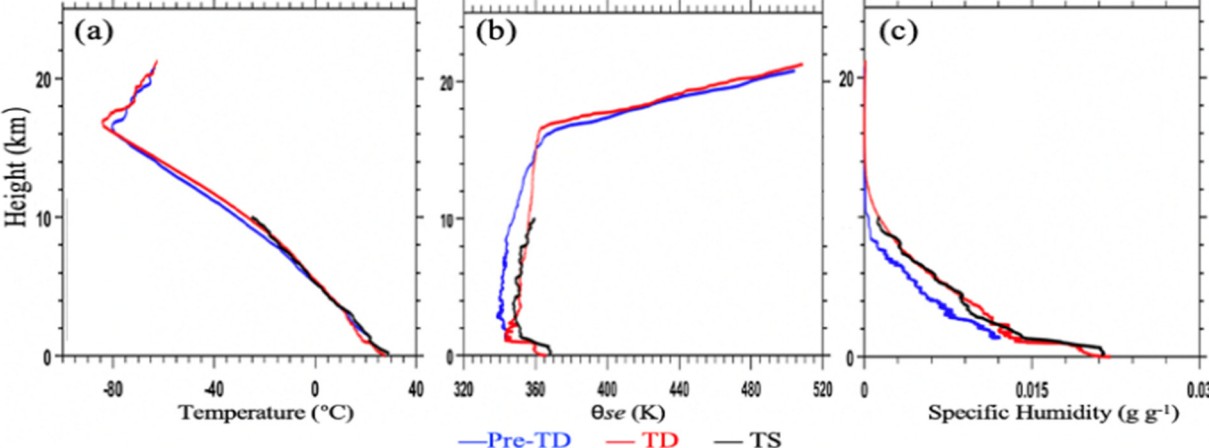

**Figure 6.** The profiles of (**a**) temperature, (**b**) pseudo-equivalent potential temperature (θ$se$), and (**c**) specific humidity at the three different evolutionary stages of Typhoon Sinlaku: (**a**) pre−TD, (**b**) TD, and (**c**) TS.

Additionally, $\theta_{se}$ in the three stages was found to decrease with the altitude below 2 km AMSL; temperature inversions occurred frequently within this altitude range, particularly in the TS stage (Figure 7c). There seemed to be a typical capping inversion that indicated a downward motion above 2 km AMSL. In addition, it was speculated that $\theta_{se}$ near the eyewall area of the typhoon was higher, and the wind speed became larger [44,78]. As a result, local $\theta_{se}$ increased. Figure 7 shows the skew T-logP diagram of the soundings in the three stages. It was found in the pre-TD stage that the humidity at the bottom was larger than that in the middle and high levels, which was favorable to the occurrence of severe convective weather like a typhoon. As it further evolved into the TD stage when Typhoon Sinlaku (2020) passed over the Xisha Islands, the profiles of temperature and dewpoint were

close to each other, starting from the sea level up to the tropopause (Figure 7b), suggesting that abundant water vapor was pervasive through almost the whole troposphere, that the atmosphere was supersaturated, or that there was cloud droplet–laden troposphere. This was well corroborated by the cyclonic cloud covering over the study area from the Himawari geostationary RGB image as shown in Figure 3e.

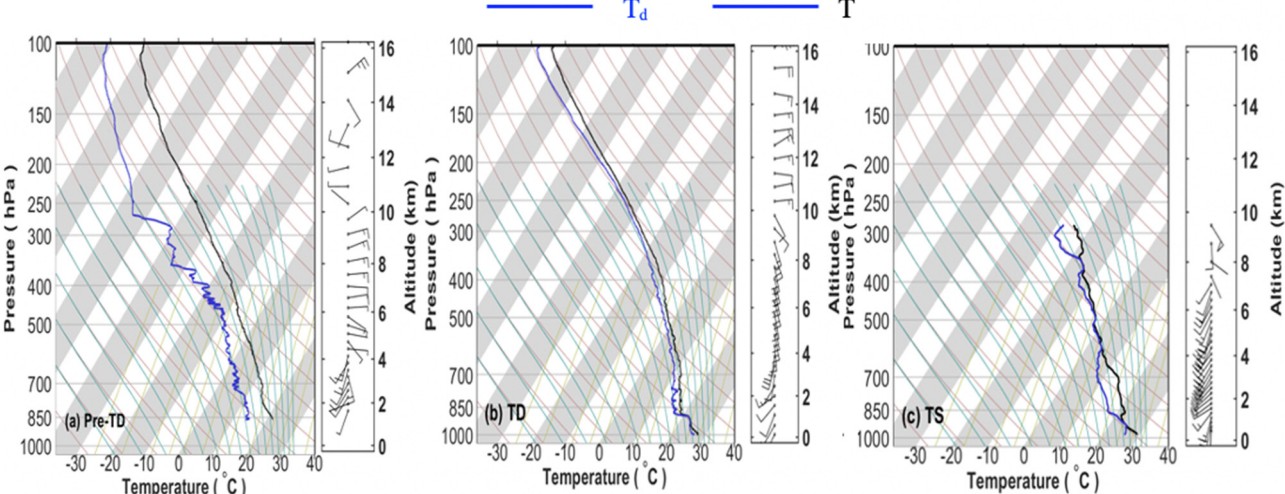

**Figure 7.** The skew T−log P diagrams from two-way rawinsonde measurements at the stages of (**a**) pre-TD and (**b**) TD and from dropsonde measurements at the stage of (**c**) TS for Typhoon Sinlaku (2020).

It is noteworthy that a saturated atmosphere in the middle troposphere and low boundary layer was found on 2 August 2020, even though Typhoon Sinlaku (2020) had passed away from the observational site at the Xisha Islands (Figure 7). At a matter of fact, Typhoon Sinlaku (2020) weakened as TS and persisted over the South China Sea for several days (Figure 3f).

## 5. Concluding Remarks

The atmospheric thermodynamic and dynamic structures of Typhoon Sinlaku (2020) at different evolutionary stages were comprehensively analyzed using the multiple observations from dropsondes, two-way rawinsondes, satellite-based sensors, and a surface-based weather station. In terms of atmospheric dynamic structure, the horizontal wind speed increased initially up to 9 km AMSL, reaching a maximum at 17 km AMSL during the pre-TD and TD stage of Typhoon Sinlaku (2020). Intriguingly, the maximum local vertical wind shear also appeared at 3 km AMSL before the initiation of the TD. However, strong turbulence existed in the upper and lower atmospheric layers as this typhoon was strengthening, leading to a decreasing trend in local vertical wind shear in the TS stage. In addition, there were strong variations in wind speed profiles between the different evolutionary stages of this typhoon. The maximum low-level wind speed during the TS stage was nearly twice the pre-TD stage, when the cyclonic circulation of the typhoon was increasing. In terms of thermodynamic structures, temperature, specific humidity, and equivalent potential temperature decreased with height in the troposphere. However, temperature inversions were frequently found below 2 km AMSL, which may be due to the higher equivalent potential temperature close to the eye of typhoon compared with other altitudes when the wind speed changed. Interestingly, the temperature profiles did not vary much in the different stages of Typhoon Sinlaku (2020). The minimum temperature lapse rate was found during the TD period, as compared to the maximum temperature lapse rate during the TS stage in the lower atmosphere. In addition, a large temperature lapse rate was found in the TD stage in the upper atmosphere. With the development of the typhoon, energy was constantly obtained from the sea surface, resulting in the increase

in specific humidity and $\theta_e$. This indicated a continuous strengthening of the typhoon to a certain extent, which provided a key reference for typhoon intensity forecasting.

Even though we gained the thermodynamic and dynamic features of Typhoon Sinlaku (2020) by investigating the profiles of temperature, wind, and humidity, our study remains far from a holistic view of the inner structure of the typhoon. In the future, more field campaigns or experiments with air-borne dropsondes, in combination with the profiling capability of meteorological satellites, are warranted for fully elucidating the fine internal dynamic and thermodynamic structure of a typhoon.

**Author Contributions:** Conceptualization, J.G. and L.L.; methodology, Y.H.; software, L.L. and Y.H.; validation, J.G. and L.L.; formal analysis, L.L. and Y.H.; investigation, L.L., Y.H. and J.G.; resources, L.L.; data curation, L.L. and Q.G.; writing—original draft preparation, L.L. and Y.H.; writing—review and editing, L.L., Y.H., Y.X., W.G., Q.G. and J.G.; visualization, Y.H.; supervision, J.G.; funding acquisition, J.G. All authors have read and agreed to the published version of the manuscript.

**Funding:** This research was funded by the Natural Science Foundation of China under grant number U2142209.

**Institutional Review Board Statement:** Not applicable.

**Informed Consent Statement:** Not applicable.

**Data Availability Statement:** The sounding data over the Xisha Islands for the Sinlaku Typhoon (2020) are publicly available at https://doi.org/10.5281/zenodo.5893813 (accessed on 18 April 2022).

**Acknowledgments:** We appreciate the high-resolution sounding measurements provided by the Meteorological Observational Center of China Meteorological Administration. We are also grateful to the China Meteorological Administration (http://tcdata.typhoon.org.cn/ (accessed on 18 April 2022)) for generously sharing the valuable typhoon best track data. Last but not least, the authors would like to thank the editor and the anonymous reviewers for their constructive comments, which helped improve the quality of our manuscript.

**Conflicts of Interest:** The authors declare no conflict of interest. The funders had no role in the design of the study; in the collection, analyses, or interpretation of data; in the writing of the manuscript, or in the decision to publish the results.

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
