# Peer review of "Investigation of Atmospheric Dynamic and Thermodynamic Structures of Typhoon Sinlaku (2020) from High-Resolution Dropsonde and Two-Way Rawinsonde Measurements"

_remotesensing, doi:10.3390/rs14112704_

Round 1

Reviewer 1 Report

Reviewer’s recommendation: Major revision

General comments:

The authors have addressed my concerns and the quality has been much improved. However, there are still many basic mistakes that should be corrected before publication.

Major comments:

  1. L273-286: This change may reflect differences in the distance from the TC center. It should be carefully remarked.

Minor comments:

  1. L6: The numbers for the authors’ institutions should be in order.
  2. L22: The term “wind shear” usually means the deep-layer vertical wind shear represented by the difference of wind between 850 hPa and 200 hPa. To avoid unnecessary misunderstanding, I recommend using the term “local vertical wind shear”.
  3. L51-54: I think that the current observations were also obtained near the coast. I wonder why this sentence is here.
  4. L58: One period is excessive.
  5. L76: This asymmetry contributing to the rapid intensification is only discussed when the translation speed is fast.
  6. L200: “dropsondes”-->”dropsondes and two-way rawinsondes”
  7. Figures: Many characters are too small to read. Please enlarge them for the reader’s sake, in particular Figures 1 and 7.
  8. The leftmost column of Table 1: It’s better to use July and August as the name of the months.
  9. L212-213: The figure caption says that Figure 2c is for two-way rawinsonde and dropsondes. However, considering the flight path shown in the right bottom of Figure 1, it does not cover the dropsondes (at least leg III). Please check the region and coverage of dataset again.
  10. L227: this typhoon is --> the lifetime of this typhoon is
  11. Figure 4: Strange white blank in the upper left corner of Figure 4a.
  12. L293: The explanation should be more elaborated. The wind speed generally decreases with height in the troposphere.
  13. L297-299: I suggest deleting the physical explanation. Matsuno (1971) did not discuss such interaction.
  14. L304: See my minor comment 2.
  15. L323: Do you mean the warming in the upper troposphere?
  16. L321-336: The observation presumably captured the air warmed by the condensation, which is a good signature of an upper air in the tropical cyclone.
  17. L333: \theta_e should be \theta_se.
  18. L334: I guess it is 0.002 g g-1.
  19. L342: I find the temperature inversion in Figure 7c. The figure number should be remarked here. It seems a typical capping inversion that indicates the downward motion above it.
  20. L347: I could not understand the meaning of “conductive.”
  21. L374-376: These features are valid for the troposphere but not the stratosphere.

Reviewer 2 Report

I want to highlight some points that on my opinion need to be revised:

  • The definition of the three phases of the storm (pre-TD, TD and TS) is given at the beginning of section 3.1, but the authors refer to the three stages also before, in two occasions in Section 2, both in the caption of Figure 2 and at the beginning of the Section. Such definition should precede each other usage of the terms.
  • Line 144. The UAV acronym is used before its definition, given (actually twice) on line 173. Maybe the phrase on line 173 can be moved at line 144.
  • The authors should specify better what two-way rawinsonde are and how they work.
  • Moreover, in Figure 3, the triangles the author refer to, are not readable. The authors can consider to make them bigger. In the figure caption, the word “dropsondes” should not be in parenthesis, since this is a little confusing, letting the reader intend that the two-way rawinsonde and the dropsonde are the same think.

Round 2

Reviewer 1 Report

The quality of the manuscript has been much improved. It is acceptable for me to recommend the publication of this in Remote Sensing. Below are minor comments for inadvertent errors. I do not need to check the revision further.

Table1: August-31 --> July 31

L201: vertical wind shear --> local vertical wind shear

Author Response

The quality of the manuscript has been much improved. It is acceptable for me to recommend the publication of this in Remote Sensing. Below are minor comments for inadvertent errors. I do not need to check the revision further.

Reply: We appreciate the invaluable comments made by the reviewer on our revised version in this round of review. Per your kind suggestion, we have made the minor revision to our manuscript. We hope the reviewer is satisfied with our revision. For clarify purpose, we highlight our replies in ltalic font as follows.

Table1: August-31 --> July 31

Reply: Amended as suggested. 

L201: vertical wind shear --> local vertical wind shear

Reply: Amended as suggested. 

This manuscript is a resubmission of an earlier submission. The following is a list of the peer review reports and author responses from that submission.

Round 1

Reviewer 1 Report

Review on “Investigation of atmospheric dynamics and thermodynamic structures of Typhoon Sinlaku from high-resolution dropsonde measurements” by Liu et al.

Recommendation: Reject

General comments:

The authors try to show the dynamic and thermodynamic structures of Typhoon Sinlaku (2022) by using two-way rawinsondes and UAV dropsondes. Since in-situ measurement of typhoon regions is limited in the western North Pacific, this work potentially has the value to be published in terms of new observation techniques. However, the understanding of the structure is very superficial, and the manuscript contains a lot of errors to list. I recommend adding more specialists on the atmospheric sciences and tropical cyclone dynamics and/or focusing on the new observation technique.

Major comments

  1. Interpretation: The dynamic and thermodynamic condition differ in the eye, eyewall, outer core, and ambient regions (e,g., Montgomery et al. 2006, BAMS). However, the current work did not consider the location of in-situ observations with respect to the storm center. I understand this storm is in developing stage and the convection is still not well organized. But the structures may reflect the different feature in different portion of the storm rather than the different stage (L221-223, L305-309, and others). The authors should not discuss the difference of stages.
  2. Insufficient references: A lot of works have been done with the dropsonde observations previously. Even if we limit the studies for TCs in the western North Pacific, Chou and Wu (2008, Monthly Weather Review) and Ito et al. (2018, SOLA) showed the improvement of forecast skill, and Yamada et al. (2021, J. Meteorol. Soc. Japan) and Hirano et al. (2022, J. Atmos. Sci.) showed the dynamical and thermodynamical features of typhoons.
  3. Insufficient understandings: There are many basic mistakes and insufficient understanding of atmospheric sciences and typhoon dynamics. Here are some examples:
  • L204-212: Something is wrong with the value of equivalent potential temperature (\theta_e). If \theta_e really equals to around 300 K, it means almost no water vapor mixing ratio. It is very strange for the thermodynamic condition.
  • L294: The vertical wind was neither measured nor estimated.
  1. L230-234: It seems the ordinary easterly wind in the stratosphere. Is there any evidence that the typhoon strengthened this?
  2. L272: The existence of the outflow was not shown with the current data.
  3. L263-265: Since the distance from the storm center differs in each observation, the authors should not conclude this.
  4. It seems there are no new scientific findings from the current work. Please explain what are new.

Minor comments

  1. The title is misleading. The analysis uses both two-way rawinsondes and dropsondes. The authors should include the two-way rawinsondes in the title. Also, Sinlaku should be written as Sinlaku in 2020 because the same typhoon name is repeatedly used every 140 TCs.
  2. L18: dropsondes --> rawinsondes and dropsondes
  3. L28: thermal --> dynamic
  4. L35: storm, and typhoon
  5. L52: observations --> in-situ observations (consider satellites and radars)
  6. L56-58: Fudeyasu et al. (2018, J. Climate) showed that the weak vertical wind shear is favorable for the rapid intensification based on large samples.
  7. L91: this research --> the research
  8. L94: slope of what?
  9. L108-116: What physical variables were measured?
  10. L128-129, L199: Why was only the third leg used? It is valuable to show and analyze all legs.
  11. Section 2: UAV dropsondes were deployed on the peripheral of the storm, while the two-way rawindondes obtained the pre-storm condition and storm-inner core. Please explain these facts.
  12. Figure 2b: What does a red asterisk indicate?
  13. Many figures: Some fonts and graphs are very small to see the values. Enlarge the fonts and make the figures inside a figure independent.
  14. Table 1: 11:51 in the descent stage seems to be 12:28.
  15. Table 1: Explain it uses LST.
  16. L233 and others: Use the superscript -1.
  17. L242-243: Which figure of Stern and Zhang?
  18. Section 2 and Figure 3: Was the observation conducted in a heavy rain condition or cloudy condition? Clarify whether each observation captures the condition inside the convective clouds or no cloud condition.

Reviewer 2 Report

This work was trying to investigate the dynamic and thermodynamic structures of a Typhoon using high-resolution dropsonde measurements. However, as written, there are lack of new insights in terms of science and technology.  I would suggest the authors spend time describe the measurements in more details (including error/uncertainty analyses) and improve their analyses of typhoon structure from the data.  

(1) The authors might need to improve their presentations, particularly language/grammar.  

(2)  Figs5/6/7.  I wondering  where (e.g. distance from center)/what time periods of measurements are used to construct the vertical profiles.  There are significant drifts with strong winds in the dropsonde/balloon measurements. (i.e., long trajectories ). How do you consider the differences in location and time?  Vertical structures could be significantly different at different locations/times of a typhoon. Need to discuss how do you obtain/screen those profiles, and how well they represent the structure.

Reviewer 3 Report

The manuscripts “Investigation of atmospheric dynamic and thermodynamic structures of Typhoon Sinlaku from high-resolution dropsonde measurements “ by Liu et al., aims at the description of the internal structure of a Tropical Cyclone. Different datasets are used for the analysis. At line 135 the author state that “The main dataset used in our study include but not limited to…”. I did not find in the text analyses with data different from that reported in lines 135-140. To which other data do the authors refer in line 135? In addition, in Figure 3, the upper panels (a, b and c) report data from the GFS reanalysis. However, GFS data are not mentioned in the comment of the figure, but only in its caption, and also in lines 135-136 is not specified at what time of the day GFS data are referred for the three days considered in the figure (maybe this time is inside the time range of the balloon measurements, but it is better to specify it more precisely). The time of the day is not specified also for TBB data. Furthermore, the manuscript highlights the importance of dropsonde measurements both in the title and in the introduction, it is then important to specify better which was the added value of the dropsonde measurements in the description of the Typhoon structure.

Some minor points are then addressed below:

Line 177 “strom” should be “storm”

Line 171-173 Table 1 e Table 2 are mentioned in Section 2 and therefore should be placed in Section 2 and not at the beginning of Section 3.

Lines 233-237-238 in “m s-1” , “-1” should be a superscript

Line 281 shown Figure 3e -> shown in Figure 3e